# OpenReview forum: "Aligning Large Multimodal Models with Factually Augmented RLHF"
_ICLR.cc/2024/Conference — Submitted to ICLR 2024_

### Official Review · Reviewer_XL3H · 2023-10-22

**Soundness:** 3 good
**Presentation:** 2 fair
**Contribution:** 3 good
**Rating:** 6
**Confidence:** 4

**Summary:**

This paper presents an innovative approach to address multimodal misalignment in Large Multimodal Models (LMM) by adapting Reinforcement Learning from Human Feedback (RLHF) to vision-language alignment. The proposed Factually Augmented RLHF method enhances the reward model with factual information, improving performance and mitigating reward hacking issues.

**Strengths:**

1. Multimodal Misalignment Addressed: This study effectively tackles the issue of multimodal misalignment in Large Multimodal Models by adapting Reinforcement Learning from Human Feedback to vision-language alignment, improving model performance.
2. Factually Augmented Reward Model: The proposed Factually Augmented RLHF method enhances the reward model with factual information, which not only improves performance but also mitigates reward hacking issues in RLHF.
3。 Performance Improvement: By training the model with augmented data and utilizing RLHF, this approach achieves remarkable performance gains.

**Weaknesses:**

1. The authors proposed that the evaluation indicates that this benchmark dataset aligns well with human evaluations, especially when scores are adjusted for anti-hallucinations. I didn’t find how to get this conclusion in the experimental section or other sections.
2. The method section requires an overall overview of the relationships between the various parts of the method.
3. Are answers and questions in the MMHAL-BENCH dataset written by humans or constructed by automatic methods?

**Questions:**

see weakness.

---

> ### Author Response · Authors · 2023-11-22
> **Reply to Reviewer XL3H**
>
> Thank you for your insightful feedback! We appreciate your recognition of our approach in tackling multimodal misalignment and enhancing RLHF with factual information. Your constructive comments of our method's effectiveness and contribution is highly valued. We address your questions below.
>
> > Concern 1: Are answers and questions in the MMHAL-BENCH dataset written by humans or constructed by automatic methods?
>
> The questions and answers are created by humans. As introduced in Appendix C, we have selected 8 question categories $\times$ 12 object topics, and manually designed questions where prior LMMs usually make false claims about image contents (hallucinations).
>
> > Concern 2: The authors proposed that the evaluation indicates that this benchmark dataset aligns well with human evaluations, especially when scores are adjusted for anti-hallucinations. I didn’t find how to get this conclusion in the experimental section or other sections.
>
> After designing the questions, we have tested LLaVA-13B and IDEFICS-80B on MMHal-Bench. We manually inspected 1) the answers produced by these two LMMs, and 2) their corresponding reviews given by GPT-4. When judging whether hallucinations exist in the LMM responses, we found that the agreement between human inspection and GPT-4 evaluation was 94% (Appendix C). It is worth noting that IDEFICS-80B was not involved when we designed the question-answer pairs, but the hallucination evaluation is still highly consistent between humans and GPT-4. These observations demonstrate that the automated GPT-4 evaluation in MMHal-Bench is well-aligned with humans.
>
> > Concern 3: The method section requires an overall overview of the relationships between the various parts of the method.
>
> We thank the reviewer for the insightful suggestion. We have added an overview paragraph in Section 2.

---

### Official Review · Reviewer_Y7nK · 2023-10-25

**Soundness:** 2 fair
**Presentation:** 3 good
**Contribution:** 2 fair
**Rating:** 5
**Confidence:** 3

**Summary:**

This paper is a vision-language paper with three contributions:
- introducing LLaVA-RLHF, a vision-language model trained via reinforcement learning from human feedback (RLHF) to improve multimodal alignment
- developing Factually Augmented RLHF to mitigate the problem of reward hacking in RLHF by leveraging additional factual information, such as image captions
- proposing an evaluation benchmark for hallucination, MMHAL-BENCH

Concretely, the paper starts with tuning the LLaVA model using existing vision-language datasets (e.g., VQA-v2 and Flickr30k). Then, the 10k preference data was collected by human annotators, selecting the preferred response among two responses. Next, the reward model for RLHF is trained on the collected preference data. Unlike the vanilla reward model that utilizes the preference data, the proposed method (i.e., factually augmented RLHF; Fact-RLHF for short) injects additional information (e.g., image captions) into the reward model. So, Fact-RLHF requires additional information in both training and inference. Finally, the instruct-tuned LLaVA model is optimized to generate the responses having maximum rewards.

The paper verifies the effectiveness of the proposed method on three existing benchmarks (MMBench, POPE, and LLaVA-B) and a new benchmark, MMHAL-B.

**Strengths:**

(S1) Application of RLHF to the vision-language domain is definitely a novel direction.

(S2) The paper conducts a wide range of experiments to demonstrate the usefulness of Fact-RLHF on diverse benchmarks.

(S3) The paper presents a new benchmark dataset, MMHAL-BENCH.

(S4) The paper is well-written and easy to understand.

**Weaknesses:**

(W1) I could NOT find direct evidence that the proposed method reduces hallucinations and mitigates reward hacking. As shown in Table 6, LLaVA-RLHF did not significantly differ from LLaVA-SFT in the hallucination rate. LLaVA-RLHF even falls behind LLaVA-SFT in the 13B model. More convincing evidence or a detailed analysis is required that Fact-RLHF truly improves hallucinations. I am not expecting the response that LLaVA-RLHF just outperforms LLaVA-SFT on human alignment benchmarks (i.e., LLaVA-Bench and MMHAL-BENCH).

(W2) Fact-RLHF works when additional human-annotation information (e.g., image captions and OK-VQA rationale) is available. It implies that we can’t use Fact-RLHF when such additional information does not exist. The paper did not study more realistic scenarios where additional information is unavailable or noisy.

(W3) The paper did NOT elaborate on experimental setup (e.g., descriptions of each dataset and task & evaluation metrics), assuming that all readers are familiar with such setup. I could find some details only for MMHAL-BENCH in the appendix.
- Descriptions of each benchmark and task (e.g., basic statistics, data format, what should the model predict?)
- Evaluation protocol for each benchmark (e.g., which evaluation metric is used and how to compute the results)

**Questions:**

Please see the weaknesses

---

> ### Author Response · Authors · 2023-11-22
> **Reply to Reviewer Y7nk (1/2)**
>
> Thank you for your helpful comments! We appreciate your recognition of the novel application of RLHF in the vision-language domain and the introduction of our MMHAL-BENCH benchmark. We address your questions below.
>
> > Concern 1: I could NOT find direct evidence that the proposed method reduces hallucinations and mitigates reward hacking.
>
> Our proposed Fact-RLHF indeed has helped the model to reduce hallucination. After investigating the evaluation by GPT-4 on MMHal-Bench, we would like to clarify that our model trained with **Fact-RLHF was actually at a disadvantage** when being evaluated with open-ended questions. The reason is that RLHF encourages the model to make detailed, informative responses, which are often significantly longer than responses given by a model only trained with SFT, as well as human annotators. Therefore, our RLHF model may include some information that are valid statements about the image, but just not included in the human-generated answer or image annotations. Consequently, GPT-4 would mistakenly believe that our RLHF model had hallucinations. We observe that such biased evaluation exists in our RLHF-based models (both 7B and 13B), but not in our model only trained with SFT or other LMMs, since they are not further tuned by RLHF.
>
> Let’s see some examples:
>
> Example 1
> - Image: https://farm1.staticflickr.com/3815/9340455006_25a885f7ef_o.jpg
> - Question: Who in the picture is wearing trousers?
> - Image content annotations: "Clothing", "Human body", "Person", "Man"
> - Human answer: The woman in the image wears the pants.
> - LLaVA-SFT+-13B answer: The woman in the picture is wearing trousers.
> - GPT-4 rating for LLaVA-SFT+13B: 5, very informative, no hallucination
> - LLaVA-RLHF-13B answer: In the picture, the woman is wearing trousers while sitting next to the man. The man is wearing shorts, and both of them are sitting in front of a campfire. This indicates that they are likely on a camping trip or enjoying an outdoor activity together. The presence of a campfire suggests that they are in a natural setting, possibly in a park or a campsite. The fact that they are both sitting in chairs around the fire implies that they are taking a break from their activities, perhaps to relax, socialize, or cook food. Their choice of clothing, with the woman wearing trousers and the man wearing shorts, suggests that they are dressed for a warm and comfortable outdoor experience. The woman's choice of trousers may provide her with more protection from the elements, such as cold temperatures or insects, while the man's shorts allow for better air circulation and freedom of movement.\n\nOverall, the image of a woman wearing trousers and a man wearing shorts sitting around a campfire represents a moment of leisure and enjoyment in the outdoors, where they can appreciate the natural beauty and serenity of their surroundings.
> - GPT-4 rating for LLaVA-RLHF-13B: 1, somewhat informative, with hallucination
> - Explanation: LLaVA-RLHF-13B correctly identifies the person “wearing trousers,” but also includes additional information that can be observed in the image. However, the additional information (such as the campfire and chairs) are not included in the human-generated answer or the annotations. As a result, GPT-4 incorrectly assigns a low score to the answer.
>
> Example 2
> - Image: https://c2.staticflickr.com/4/3501/4068897399_a729022693_o.jpg
> - Question: Describe the image in detail.
> - Image content annotations: "Window", "Clothing", "Person", "Building"
> - Human answer: The image captures a bustling street lined with various shops and an adjoining sidewalk. Several pedestrians are present, with two individuals notably passing a store with a green sign saying "Hawkin's Bazaar". At this store's entrance, a man stands, overseeing a spacious window display that houses diverse items, highlighted by a red and white sign. Interestingly, no trees or parked cars are present in the street. The mood is energetic, showcasing individuals engaged in their daily routines and shopping pursuits.
> - LLaVA-RLHF-13B answer: The image depicts a lively street scene with a group of people walking past a store called "Hawkins Bazaar." The storefront is adorned with Christmas decorations, and the store appears to be selling a variety of items, including toys and gifts. There are several people walking along the sidewalk, with some of them carrying handbags. The pedestrians seem to be enjoying their time, browsing the storefronts and engaging in conversation. The street is bustling with activity, and the store is likely attracting customers with its festive decorations and diverse selection of products.
> - GPT-4 rating for LLaVA-RLHF-13B: 1, somewhat informative, with hallucination
> - Explanation: Similarly in this example, the model mentions additional information about “Christmas decorations,” which is valid but not included in the annotations or human-generated answer. Therefore, GPT-4 fails to correctly assess the response.
>
> (cont.)

---

> ### Author Response · Authors · 2023-11-22
> **Reply to Reviewer Y7nk (2/2)**
>
> (cont.)
>
> Due to limited space, we cannot include too many examples in this response, but we will include more detailed analysis regarding the under-estimation of RLHF models’ performance in the manuscript revision.
>
> To summarize, RLHF would enrich our LMM’s responses. However, informative and detailed responses by RLHF models sometimes cannot be matched with elements in the annotations and human-generated answers. Although we have tried our best to make human answers as comprehensive as possible, there could always be more to include. As a result, the hallucination rate of our RLHF model is overly estimated by GPT-4. If we take this inherent bias into account, RLHF is indeed reducing the hallucinations.
>
> > Concern 2: The paper did not study more realistic scenarios where additional information is unavailable or noisy.
>
> Fact-RLHF only needs question or image related factual information to augment the reward model. It is relatively cheap to annotate such factual information [1][2]. We also expect automatic caption generation techniques such as CapFilt [3] or DSC [4] (i.e., from a different model) can help automate this process.
>
> > Concern 3: The paper did NOT elaborate on experimental setup (e.g., descriptions of each dataset and task & evaluation metrics)
>
> Thanks for pointing this out. We have previously included the implementation details as well as the hyperparameters in Appendix I.
>
> Regarding the description of each dataset, task and metrics, we have added more information in Appendix. Specifically, the three employed supervised fine-tuning datasets are VQA-v2, AK-VQA and Flickr30k as listed in Section 2.2.  We use “Yes” or “No” queries from VQA-v2 (83k), multiple-choice questions from A-OKVQA (16k), and grounded captions from Flickr30k (23k). The 10k human preference data are paired outputs from the base 7B LLaVA model and we ask the Amazon Turker annotators to label which one contains less hallucinations. The details about the collection process is in Appendix G.
>
>
> For each evaluation task, we report the accuracy for MMBench, which is a multiple-choice question benchmark consisting of 1031 questions. We report the F1 score for the POPE, which is a “Yes/No” question benchmark and consists of 3k questions in three categories (random, adversarial and popular).
> The LLaVA bench consists of 100 questions and will be evaluated by GPT4 against the outputs from text-only GPT4. Finally, we report the GPT4 score on MMHalBench, which has 96 questions as well targeting the hallucination level of each model.
>
>
> [1] Chen, Xinlei, et al. "Microsoft coco captions: Data collection and evaluation server." arXiv preprint arXiv:1504.00325 (2015).
>
> [2] Pont-Tuset, Jordi, et al. "Connecting vision and language with localized narratives." ECCV, 2020.
>
> [3] Li, Junnan, et al. "Blip: Bootstrapping language-image pre-training for unified vision-language understanding and generation." International Conference on Machine Learning. PMLR, 2022.
>
> [4] Betker, James, el al., “Improving Image Generation with Better Captions”

---

> > ### Comment · Reviewer_Y7nK · 2023-11-23
> > **Thanks for your response**
> >
> > I want to thank the authors for responding to my concerns. I have follow-up comments about the authors' responses.
> >
> > For the first concern, the authors explained the reason for the disadvantage of Fact-RLHF in the current GPT-4-based evaluation. These two examples show that the proposed method (i.e., Fact-RLHF) tends to generate detailed responses to the questions. Unfortunately, it seems difficult for me to be convinced that Fact-RLHF mitigates hallucinations. This is because the responses from Fact-RLHF still include informative but unnecessary explanations about the given question. For this reason, **a human study** should be conducted to address this kind of concern.
> >
> > For the second concern, the models the authors mentioned (i.e., CapFilt [3] or DSC [4]) sometimes provide **noisy or incorrect information**. So, how would Fact-RLHF generate the responses when **the golden** factual information is not given?

---

### Official Review · Reviewer_vv7w · 2023-10-28

**Soundness:** 3 good
**Presentation:** 3 good
**Contribution:** 3 good
**Rating:** 6
**Confidence:** 4

**Summary:**

This work performs RLHF for the vision-language alignment of LMM models. This work proposes a new alignment algorithm called Factually Augmented RLHF to reduce the hallucination in vision-language tasks. It also contributes a new benchmark named MMHAL-BENCH to evaluate LMM’s performance with a focus on hallucinations.

**Strengths:**

1. This work is one of the first applications of RLHF to improve the multimodal alignment of LMMs. This is an inevitable next step for the LMM research.

2. It proposes a new hallucination benchmark named MMHAL-Bench.

3. This work shows that RLHF indeed improves the performance of the LLaVA model in LLaVA-Bench and MMHAL-Bench.

**Weaknesses:**

1. The key issue of this work may be lack of details, as the method section only covers high-level procedures.
- RLHF is notoriously difficult to reproduce.
- No information is given about labelers, such as how many annotators participate in, who they are, and how they are recruited, etc.

2. More experimental results are desired, as this work reports the final performance only.
- Is there any quantitative evaluation that focus on how much hallucination reduces?
- It would be better to include some experimental results about the performance of the reward model.

3. The supplementary file is useless for reviewing. Only a github link is provided, but it is not accessible.

**Questions:**

1. FACT-RLHF is sourced from QA pairs in the LLaVA, A-OKVQA and VQA-v2. Are they any other datasets to get more samples?

2. How much does it cost to collect FACT-RLHF annotations? How long does it task? How many labelers are involved?

**Details Of Ethics Concerns:**

This work exploits large-scale human feedback to learn AI models, but much details are not shown in the draft. Thus, it is not possible for reviewers to understand any ethical issues that this work may involve.

---

> ### Author Response · Authors · 2023-11-22
> **Reply to Reviewer vv7w**
>
> Thank you for your helpful comments! We appreciate your recognition of the novelty and significance of our work in applying RLHF to improve multimodal alignment in LMMs, as well as your acknowledgment of the importance of our newly proposed MMHAL-Bench. Your appreciation of these aspects validates the direction and relevance of our research. We address your questions below.
>
> > Concern 1: The key issue of this work may be lack of details
>
> > Concern 1: The supplementary file is useless for reviewing
>
> We have updated the supplementary material to share all the details of our implementation. We will also open-source the code, model, and the collected datasets.
>
> > Concern 2: No information is given about labelers
>
> > Question 2: How much does it cost to collect FACT-RLHF annotations? How long does it task? How many labelers are involved?
>
> We have added more information about the anonymized labelers in the Appendix G of the paper. We spend around $5000 to collect the annotations. It took around 1 a week on the MTurk platform, and 28 labelers were involved.
>
> > Concern 3: Is there any quantitative evaluation that focus on how much hallucination reduces?
>
> As previous LLM hallucination benchmarks are usually limited to the yes/no questions, in this paper, we mainly quantitatively evaluate the hallucination of LLaVA-RLHF with our proposed MMHAL-BENCH dataset. Please see detailed description of the MMHAL-BENCH in Appendix C and the results in Table 6. LLaVA-RLHF can improve by 60% over other baselines, and reduce the hallucination rate of the original LLaVA by 27%.
>
> > Concern 4: It would be better to include some experimental results about the performance of the reward model.
>
> We have added more information about the training and evaluation of the reward model.
>
> > Question 1: FACT-RLHF is sourced from QA pairs in the LLaVA, A-OKVQA and VQA-v2. Are they any other datasets to get more samples?
>
> LLaVA-1.5 [1] is trained on LLaVA, A-OKVQA, VQA-v2, and many more datasets like TextCaps, RefCOCO, and VG. LLaVA-RLHF can also potentially benefit from those additional vision-language alignment data.
>
>
> [1] Liu, Haotian, et al. "Improved baselines with visual instruction tuning." arXiv preprint arXiv:2310.03744 (2023).

---

### Official Review · Reviewer_Kjps · 2023-10-31

**Soundness:** 3 good
**Presentation:** 1 poor
**Contribution:** 2 fair
**Rating:** 3
**Confidence:** 3

**Summary:**

## Summary
- Introduces a new multimodal large language model - LLaVA-RLHF for improved multimodal alignment. Make a variety of enhancements at various stages of the pipeline.
	- Instruct Tuning stage:
		- Augments the vision synthetic instruction tuning dataset (LLaVA dataset 98k conversations + 60k conversations for preference modeling) with human annotated multi-modal data in conversation format by using the VQA-v2, A-OKVQA dataset (converted to multi-round QA task) and Flickr30K dataset (converted to spotting captioning task) to train LLaVA-SFT+ model
	- Preference Modeling:
		- They collect preference dataset by creating a set of varied questions covering 12 main object categories from COCO and 8 different task types. This is released as MMHAL BENCH
		- This human preference dataset is collected for 10K held out examples from LLaVA
	- RL Tuning:
		- Finally they using RL tuning where they propose a new alignment algorithm called Factually Augmented RLHF for improved multimodal alignment.
			- Reward model training
				-  The key idea in Fact-RLHF is to use additional information from existing datasets (e.g. image captions from the coco dataset/rationales from A-OKVQA dataset). This additional factual data is provided to the reward model both at training and inference time.
			- RL Tuning
				- RL tune using PPO on the 10K held out examples collected from LLaVA
- The model improves by 96% LLaVA-Bench dataset  compared to text-only GPT-4 which is better than 87% improvement of previous methods.
- The model improves by 60% on the MMHAL-BENCH benchmark developed in the paper.

**Strengths:**

## Strengths/Weakness

- This is the first LMM trained with RLHF
- Gets SOTA results for LLaVA-Bench and MMHAL-BENCH
- The RLHF model degrades slightly on the capability benchmarks. Can you please cite the reference to the scaling law of LLaVA-RLHF mentioned in the paper?

**Weaknesses:**

## Strengths/Weakness

- This is the first LMM trained with RLHF
- Gets SOTA results for LLaVA-Bench and MMHAL-BENCH
- The RLHF model degrades slightly on the capability benchmarks. Can you please cite the reference to the scaling law of LLaVA-RLHF mentioned in the paper?

**Questions:**

## Questions/Clarifications

- Slightly confused by the data used for RLHF-RL Tuning. AFAIU the authors collect human preferences on 10k hold out LLaVA dataset. But how do you get the factual information for these examples (captions or rationals) for these required by the reward model?
- Can you please clarify how is the image captioning data from COCO converted to instruction tuning data and is it also used for RL Tuning?

---

> ### Author Response · Authors · 2023-11-22
> **Reply to Reviewer Kjps**
>
> Thank you for your comments! We appreciate your recognition of the significance of our work as the first LMM trained with RLHF and its state-of-the-art results on LLaVA-Bench and MMHAL-BENCH. We are glad that these key aspects of our research were well received and appreciated. We address your questions below.
>
> > Clarification 1: “They collect preference dataset by creating a set of varied questions…”
>
> The MMHal is only used for evaluating the hallucination in LMM generations.
>
> > Clarification 2: “The model improves by 96%...”
>
> Our method achieves an absolute performance of 96% of text-only GPT-4, not improves by 96%.
>
> > Concern 1: The RLHF model degrades slightly on the capability benchmarks.
>
> While there is a slight degradation at the 7B scale, the RLHF significantly improves performance on the MMBench at the 13B scale.
>
> > Concern 2: Can you please cite the reference to the scaling law of LLaVA-RLHF mentioned in the paper?
>
> We have added citations to the alignment scaling laws of [1,2].
>
> > Question 1: how do you get the factual information for these examples
>
> Since the images are from the MS COCO dataset, they come with 5 captions for each image.
>
> > Question 2: clarify how is the image captioning data from COCO converted to instruction tuning
>
> We did not use captioning data for instruction tuning. It is only used as factual augmentation for the reward model during the RL phase.
>
> [1] Askell, Amanda, et al. "A general language assistant as a laboratory for alignment." arXiv preprint arXiv:2112.00861 (2021).
>
> [2] Kaplan, Jared, et al. "Scaling laws for neural language models." arXiv preprint arXiv:2001.08361 (2020).

---

### Author Response · Authors · 2023-11-22
**Thank you and we are looking forward to your post-rebuttal feedback!**

Dear AC and all reviewers:

Thanks again for all the insightful comments and advice, which helped us improve the paper's quality and clarity.

The discussion phase is about to end soon and we kindly remind the post-rebuttal responses.

We would love to convince you of the merits of the paper. Please do not hesitate to let us know if there are any additional experiments or clarification that we can offer to make the paper better. We appreciate your comments and advice.

Best,

Author

---

### Meta-Review · Area_Chair_gxRr · 2023-12-10

**Metareview:**

This paper presents a method to leverage RLHF to improve multimodal LLM  (mLLM) based on LLaVA. The authors highlight the factuality alignment of the method, aiming for reducing the hallucination of the mLLMs. The RL part includes a newly collected human preference dataset, carefully designed reward model, and a new benchmark MMHAL-BENCH to evaluate the hallucination of a model.

Pros (from reviewers):
1. First RLHF work for multimodal large language model.
2. A new benchmark  MMHAL-BENCH for evaluating the hallucination of a model.
3. SoTA results on LLaVA-BENCH and MMHAL-BENCH

Cons (from reviewers):
1. Reproducibility, details of experiments and evaluations are missed in the paper.
2. Lack of human study of the final output.

I consider the second concern regarding the absence of a human study to be a significant issue. Our laboratory conducted a side-by-side comparison of various recent multimodal Large Language Models (LLMs), including LLaVA-RLHF. We observed that the outputs of the LLaVA-RLHF model are frequently excessively lengthy, and sometimes ignore the original intent of the posed question.

There may be a substantial flaw in either the human annotation process or the design of the reward model. A more meticulous comparison and investigation are necessary before the paper can be considered for publication.

**Justification For Why Not Higher Score:**

See the meta review.

**Justification For Why Not Lower Score:**

N/A

---

### Decision · Program_Chairs · 2024-01-16

Reject